# Acceptance of a Text Messaging Vaccination Reminder and Recall System in Malaysia’s Healthcare Sector: Extending the Technology Acceptance Model

**DOI:** 10.3390/vaccines11081331

**Published:** 2023-08-06

**Authors:** Kamal Karkonasasi, Yu-N Cheah, Mogana Vadiveloo, Seyed Aliakbar Mousavi

**Affiliations:** 1School of Computer Sciences, Universiti Sains Malaysia, USM Penang 11800, Malaysia; pouyaye@gmail.com; 2Faculty of Information and Communication Technology, Universiti Tunku Abdul Rahman, Kampar 31900, Malaysia; moganav@utar.edu.my

**Keywords:** text messaging vaccination reminder and recall system, technology acceptance model, nurses’ attitudes and intentions, nurses’ acceptance, artificial neural network

## Abstract

Malaysian healthcare institutions still use ineffective paper-based vaccination systems to manage childhood immunization schedules. This may lead to missed appointments, incomplete vaccinations, and outbreaks of preventable diseases among infants. To address this issue, a text messaging vaccination reminder and recall system named Virtual Health Connect (VHC) was studied. VHC simplifies and accelerates immunization administration for nurses, which may result in improving the completion and timeliness of immunizations among infants. Considering the limited research on the acceptance of these systems in the healthcare sector, we examined the factors influencing nurses’ attitudes and intentions to use VHC using the extended technology acceptance model (TAM). The novelty of the conceptual model is the incorporation of new predictors of attitude, namely, perceived compatibility and perceived privacy and security issues. We conducted a survey among 121 nurses in Malaysian government hospitals and clinics to test the model. We analyzed the collected data using partial least squares structural equation modeling (PLS-SEM) to examine the significant factors influencing nurses’ attitudes and intentions to use VHC. Moreover, we applied an artificial neural network (ANN) to determine the most significant factors of acceptance with higher accuracy. Therefore, we could offer more accurate insights to decision-makers in the healthcare sector for the advancement of health services. Our results highlighted that the compatibility of VHC with the current work setting of nurses developed their positive perspectives on the system. Moreover, the nurses felt optimistic about the system when they considered it useful and easy to use in the workplace. Finally, their attitude toward using VHC played a pivotal role in increasing their intention to use it. Based on the ANN models, we also found that perceived compatibility was the most significant factor influencing nurses’ attitudes towards using VHC, followed by perceived ease of use and perceived usefulness.

## 1. Introduction

Timely and comprehensive childhood immunization is crucial in eliminating vaccine-preventable diseases, which significantly increase mortality and morbidity rates among children under five [1,2,3,4]. According to recent data published in 2022 by the World Health Organization (WHO), Malaysia’s childhood immunization rates are higher than the global average. For example, the BCG immunization rate among infants in Malaysia was 97.66%, compared to the global average of 84%. However, immunization rates against vaccine-preventable diseases have decreased in Malaysia in recent years, particularly for follow-up doses [5]. This has led to outbreaks of diseases like tuberculosis. In 2022, there were 25,391 reported tuberculosis cases, compared to 21,727 a year earlier—an increase of 17%. The number of deaths also increased by 12%, from 2288 to 2572 during the same period [6].

Consequently, the Malaysian healthcare system still lags behind the WHO’s Immunization Agenda 2030 (IA2030) which envisions “a world where everyone, everywhere, at every age, fully benefits from vaccines to improve health and well-being” [7]. The recent decreases in immunization rates and corresponding increases in vaccine-preventable diseases like tuberculosis demonstrate that Malaysia has yet to fully align with the WHO’s goals for comprehensive immunization coverage. More efforts are needed to boost immunization rates and ensure that infants are protected against contagious illnesses. 

The incomplete immunization of infants has been attributed to parental forgetfulness about vaccination milestones, parents’ hesitancy toward childhood vaccination, and the absence of continuous and effective communication between parents and healthcare organizations [3,8,9]. Parental forgetfulness may be due to long breaks between immunization appointments and the overwhelming demands of work and household responsibilities. Parents’ hesitancy to vaccinate their children is often caused by misconceptions about vaccine safety propagated by unreliable internet sources and mass media [9]. Moreover, inefficient communication may stem from the time-consuming and resource-intensive nature of paper-based vaccination systems, which are commonly used by Malaysian healthcare institutions [8]. These systems require staff to manually schedule appointments and remind parents, which is unproductive [1].

We study Virtual Health Connect (VHC) [1], a text messaging vaccination reminder and recall system that aims to replace the current inefficient paper-based systems in Malaysia’s healthcare sector. VHC aims to improve the completion and timeliness of immunizations among children using VHC by reminding and recalling parents about their infants’ immunization milestones, educating them about the importance of vaccination for their infants’ immunity against vaccine-preventable diseases, and simplifying and accelerating the immunization administration process for nurses. The system consists of two main components: a web-based application and a Short Message Service (SMS) application. Figure 1 illustrates how the VHC manages vaccination schedules. When parents bring their children to healthcare organizations for the first vaccination, nurses register their information using the web-based application. The system then schedules the upcoming immunization appointments and sends SMS reminders in English and Bahasa Malaysia to parents’ cellphones two days before each appointment. If the child receives the vaccination on time, the system automatically schedules the next appointment and sends a reminder. Otherwise, the system reschedules the overdue vaccination and recalls the parents two days before the new date. This process continues until the child is fully immunized.

VHC relies on SMS to remind and recall parents about their infants’ immunization appointments. This service has shown a remarkable capacity for illness management [2] and increasing attendance for health services [3,10] in developing countries since the majority of families are of lower socio-economic status and commonly use low-cost cellphones with text-only plans. The use of SMS reminders has effectively increased vaccination coverage for various immunization programs worldwide [2,8,11,12,13].

A review of the literature shows that several studies, including those by Ehlman et al. [14], Ekhaguere et al. [15], Kagucia et al. [16], MacDonald et al. [10], and Mekonnen et al. [17], have predominantly used randomized controlled trials to examine the effectiveness of SMS reminders and recalls in improving childhood vaccination coverage. However, there has been limited research on the acceptance of SMS reminders and recalls in the healthcare sector. For example, in a previous study by Karkonasasi et al. [18], a conceptual model based on the technology acceptance model (TAM) was proposed to determine the factors influencing the intention to use such a system among Malaysian health centers, but the model was not tested.

Moreover, Tagbo et al. [19] examined the acceptability of reminder and recall in vaccination services and the challenges to their implementation among service providers in Nigeria. Hofstetter et al. [20] also generally studied provider preferences and concerns regarding text message reminder and recall services for early childhood vaccinations in the USA. However, these studies primarily used descriptive statistics to examine the acceptability of SMS reminders and recalls in the healthcare sector, which lack statistical power and cannot examine the causal relationships between predictors and acceptance. 

Therefore, we aim to extend the limited research on the acceptance of SMS immunization reminder and recall systems in the healthcare sector by proposing a research model based on an extended TAM. The proposed model examines the factors affecting nurses’ attitudes and intentions to use the system. Our proposed model is novel in that it includes new predictors of attitude, namely, perceived compatibility and perceived privacy and security issues. We focused on nurses’ perspectives on the VHC because they are the primary frontline staff in healthcare institutions who are responsible for managing vaccination records [21,22]. By examining their viewpoints, we can gain insight into the factors that influence or hinder their acceptance of VHC.

We conducted a survey among nurses in Malaysian government hospitals and clinics. We analyzed the survey data using partial least squares structural equation modeling (PLS-SEM) to determine the significant factors influencing their attitudes and intentions to use VHC. In addition, we used artificial neural networks (ANNs) to determine the most significant predictors of nurses’ attitudes and intentions with higher accuracy. As a result, we can offer valuable insights to healthcare decision-makers, helping them to make informed decisions that advance health services. 

In the following sections, we present our proposed research model and develop our hypotheses. We then describe our research methodology, including survey design, sampling and data collection procedures, data examination, and the demographic profile of the respondents. We also check the collected data for non-response bias and common method bias in the research methodology section. Our results section covers the assessment of the measurement model and the structural model. We also apply a multi-analytical approach using PLS-SEM and ANNs in the results section. We then discuss our findings and mention our practical and theoretical contributions. Finally, we address the study limitations and future directions before concluding the paper.

## 2. The Proposed Research Model

The TAM, proposed by Davis [23], is a widely recognized model for analyzing the factors that influence user behavior and the use of numerous technologies. Despite being proposed over three decades ago, the model remains relevant and is still being applied, modified, and extended by researchers in several domains, including healthcare [24]. According to the TAM, a user’s attitude toward behavior and subjective criteria determine their behavioral intention towards technology, which in turn impacts their behavior. While perceived ease of use and perceived usefulness are two main factors affecting users’ information technology (IT) acceptance, the external variables of the TAM have not been clearly defined [25,26,27]. Therefore, we attempt to address the limitations of TAM by proposing new predictors of attitude, namely, perceived compatibility and perceived privacy and security issues. We also include the original predictors of TAM, namely, perceived ease of use and perceived usefulness, as they are commonly supported by the literature as discussed below. Studying the new predictors, along with the original predictors of the TAM, can provide a deeper understanding of the acceptance mechanism of VHC by the nurses. The proposed research model is illustrated in Figure 2, and the hypotheses are discussed in the following section. 

## 3. Hypotheses Development

The compatibility of VHC is conceptualized as the extent to which the system is perceived to be compatible with the current values and prior experiences of nurses, and their requirements in the workplace in administrating childhood vaccinations [28]. Nurses’ attitude toward using VHC is also conceptualized as their positive or negative feelings about working with the system [29]. Kuo et al. [30] reported that the compatibility of mobile EMR systems with nurses’ work practices enhanced their willingness to use these systems. By contrast, a lack of integration of an electronic health system with healthcare practices may lead to its failure because the users hesitate to utilize it to complete their daily work routines [31]. 

To streamline the management of childhood vaccination, VHC replaces the paper-based forms that nurses presently use with the equivalent electronic forms. Additionally, the system is easy to install, maintain, and operate on electronic devices in the workplace. Consequently, we propose that when nurses assume that VHC is compatible with their work settings, they feel optimistic about using it:

**H1:** 
*Perceived compatibility of VHC with the workplace will have a significant and positive effect on nurses’ attitudes toward using the system.*


The perceived privacy and security issues of VHC refer to the extent to which nurses believe that using the system could compromise the privacy and security of children’s and parents’ personal information [32]. These concerns may arise due to the potential for unauthorized access to sensitive information, which could result in legal issues for health centers and a loss of trust among parents who share their information with these centers [33,34]. These concerns also demonstrated a negative influence on the willingness to implement electronic medical records (EMRs) around the world, as mentioned by Jimma and Enyew [34]. Therefore, we propose that when nurses perceive privacy and security issues with VHC, their attitudes toward using the system are influenced negatively:

**H2:** 
*Perceived privacy and security issues with VHC regarding the personal information of parents and their children will have a significant and negative effect on nurses’ attitudes toward using the system.*


The perceived usefulness and the perceived ease of use of VHC refer to the degrees to which nurses believe that using the system would improve their job routine and would be effortless in administrating childhood vaccinations, respectively. Moreover, the intention to use the system is conceptualized as the degree of nurses’ intention to use VHC [29]. The previous studies on electronic health records (EHRs) [31,35,36], EMRs [37,38] and mobile EMRs [30] supported the significant relationship between perceived usefulness and intention. However, Gagnon et al.’s [35] integrated and psychosocial models did not support this relationship. In addition, the studies on EMRs [38] and EHRs [33] concluded that the perceived usefulness and the perceived ease of use significantly influenced attitudes toward using the systems. 

We assume that nurses would find VHC useful and easy to use in their workspace since it enables them to efficiently and smoothly administer immunization by using an automated SMS feature to remind and recall parents through cellphones. Moreover, they assume the system to be useful and easy to use due to its ready-to-apply vaccination plans and visual reporting tools. Therefore, we hypothesize that the perceived ease of use determines their attitude while the perceived usefulness determines their attitude and intention:

**H3:** 
*Perceived ease of use of VHC in the workplace will have a significant and positive effect on nurses’ attitudes toward using the system.*


**H4:** 
*Perceived usefulness of VHC in the workplace will have a significant and positive effect on nurses’ attitudes toward using the system.*


**H5:** 
*Perceived usefulness of VHC in the workplace will have a significant and positive effect on nurses’ intentions to use the system.*


The significant effect of attitude on intention is supported by earlier studies on hospital information technologies [21,22], EMRs [38], and mobile EMRs [39]. Considering the compatibility of VHC with the current work settings and its practical features that simplify immunization administration, nurses may feel positive working with the system, which subsequently leads to their higher intention to use it in the workplace. Therefore, we hypothesize that: 

**H6:** 
*Nurses’ attitudes toward using VHC will have a significant and positive effect on their intentions to use the system.*


## 4. Research Methodology

In the following, we will investigate the design of the survey. We then explain the sampling and data collection procedures. Next, we discuss the data examination procedure and present the demographic profile of the respondents. Finally, we examine the data for the presence of non-response bias and common method bias (CMB).

### 4.1. Survey Design

In the introduction of the survey, we explained to healthcare institutions that the survey should be completed by a randomly selected nurse who is competent in English and IT literate. The nurses were selected to complete the survey because their viewpoints about VHC would be critical to its effective acceptance, considering that they would be the main frontline staff in healthcare institutions [21,22] in managing vaccination records using VHC. Each nurse who participated in the survey represented a government hospital or clinic since we aimed to study the acceptance of VHC at an organizational level. 

In the introduction, we also briefly mentioned the advantages of VHC for simplifying and accelerating vaccination administration by healthcare institutions. The objective of the survey was to determine the factors influencing the acceptance of VHC. To provide the respondents with a clear picture of the system when completing the survey, we added some snapshots of the system, as shown in Appendix A. Participation in the survey was voluntary, and we assured participants that their data would remain confidential and would be used only for academic purposes. Several questions were then asked to understand the demographic profile of the respondents. Next, we used a five-point Likert scale ranging from “strongly disagree” to “strongly agree” to measure the measurement constructs using closed-ended questions. We adapted the measurement items from the studies in Table 1.

Two experts from the School of Management, Universiti Sains Malaysia thoroughly examined the survey for content validity to ensure that our survey covered all the relevant aspects of the constructs it was intended to measure [44]. We then refined the survey based on their feedback to avoid ambiguity and further clarified the instruments to establish the content validity of our survey. This involved changing the wording and structure of the questions to ensure that the questions were clear and easily understood by the participants. We also conducted a pilot study among 31 respondents selected using convenience sampling to examine the reliability of the measurement items. We then improved the questions and layout of the survey as some respondents found it vague. In terms of the reliability of measurement items, there were no major issues based on Nunnally [45]. We did not consider the data from the pilot study for the primary data analysis and did not invite those who participated in the pilot study to take part in the main survey to maintain the validity of the results.

### 4.2. Sampling and Data Collection Procedures

According to Malaysia, P.R.K.K. [46], Peninsular Malaysia, i.e., excluding Sabah and Sarawak, consists of 93 government hospitals and 2411 government clinics. We determined the sample size before collecting data using power analyses considering the background of the model and the data characteristics [47] since it was impractical to conduct our survey among all healthcare institutions. The smallest sample size based on Cohen [48] is 113, considering 5% as the significance level for the statistical power of 80% and R^2^ values of at least 0.1. We applied simple random sampling to give each item of the population the same opportunity to participate in the study. Considering the response rate of 25% of the pilot study, we distributed the self-administrated questionnaires in English by post among 452 government hospitals and clinics in Peninsular Malaysia with a valid address in the list of hospitals and clinics published by the Ministry of Health, Malaysia. We did not send our surveys to all government hospitals and clinics in Peninsular Malaysia due to time and budget constraints. Given the similarities between government hospitals and clinics in Malaysia, it can be inferred that our findings may have a high degree of generalizability. The survey was conducted from September 2022 to April 2023.

### 4.3. Data Examination

After receiving 128 out of 452 questionnaires, we carefully examined the data for missing values and straight-line response patterns. We removed seven cases due to the following reasons: two cases had more than 15% missing data, three respondents did not answer most items related to the attitude and intention constructs, and two cases showed straight-line response patterns. We also developed box plots using IBM SPSS Statistics 28 and found no outliers, using 3.0 as the interquartile range rule multipliers. Additionally, our analysis showed no major issues with data distribution as the absolute values of skewness and kurtosis for each item were close to 1.

### 4.4. Demographic Profile

Based on the demographic profile presented in Table 2, most of the informants were young adults between the ages of 18 and 44 (76%), with the most highly represented being those between the ages of 25 and 34 (33.1%). Additionally, the majority of the participants were female (67.8%). In terms of education, the informants with a diploma or secondary school education were the most represented (77.7%), while those with a master’s degree were the least represented (9.1%). Moreover, when it comes to working experience, the majority of those surveyed had 3 to 5 years of experience (43.8%). Finally, most participants had not previously accepted a text messaging vaccination reminder and recall system in the workplace (75.2%).

### 4.5. Non-Response Bias

Non-response bias occurs when the responses of survey participants differ significantly from those who do not respond. This can limit the accuracy of the survey analysis [49]. To detect the presence of non-response bias in our collected data, we compared the responses of the first and last 50 participants, who were considered as early and late respondents, respectively. This is because the late respondents may have a similar response pattern to those who did not participate in the survey [50]. We used an independent sample *t*-test to analyze the mean differences of all constructs for these two groups. Our results showed that the mean differences were not statistically significant. Therefore, we concluded that non-response bias was not present in our data.

### 4.6. Common Method Bias

Common method bias (CMB) is a type of bias that arises from the measurement method rather than the constructs being measured [51]. It can lead to incorrect conclusions about the relationships between constructs by either inflating or deflating the findings [52]. Therefore, it is important to check for CMB in survey studies that use self-administered questionnaires, especially when both predictor and criterion constructs are measured using items answered by the same respondents [51].

We used the procedural remedies and statistical controls proposed by Podsakoff et al. [51], Podsakoff et al. [53], and Williams et al. [54] to check for CMB. In terms of procedural remedies, we assured the respondents that their responses would be strictly confidential and that there would be no desired or correct answer. Moreover, we induced a time lag between the measurement items by inserting the demographic questions. 

In terms of statistical control, we applied Harman’s single-factor test to check for CMB. After entering all measurement items, we performed an exploratory factor analysis with an orthogonal rotation of Varimax and principal component analysis. The largest variance explained by a single factor was 28.75%. Moreover, we detected six factors that explained 66.38% of the variance, considering the Eigenvalue greater than one criterion. Therefore, CMB was not present in our data because a single factor did not explain the highest variance [51].

## 5. Results

We examined the measurement model using SmartPLS 4 software [55] and evaluated the direct effects in the structural model. Next, we identified the most important factors influencing attitude by using its significant predictors as input neurons for artificial neural networks built with SPSS 26.

### 5.1. Measurement Model

We used PLS-SEM for data analysis because it does not require a large or normally distributed sample [56,57]. We assessed the reflective measurement model’s convergent and discriminant validity using confirmatory factor analysis. Convergent validity was evaluated through factor loading, composite reliability (CR), and average variance extracted (AVE) [58].

For factor loading, we dropped one item from each of the perceived privacy and security issues and intention constructs because of outer loadings below 0.7. Although one item of the perceived ease of use construct obtained outer loadings below 0.7, we maintained the item because the construct’ CR was decreased after the removal of the item. All constructs had CR values ranging from 0.822 to 0.901, exceeding the recommended threshold of 0.7. Therefore, all constructs were reliable in terms of internal consistency. Moreover, all constructs had AVE values ranging from 0.536 to 0.714, exceeding the recommended threshold of 0.5. Therefore, convergent validity was established for the measurement model.

We examined discriminant validity using the heterotrait–monotrait ratio (HTMT). We dropped one item of the perceived privacy and security issues construct due to the high correlations with the intention construct’s items. Then, the confidence interval excluded the value 1 for all pairs of constructs. Therefore, discriminant validity was established. Table 3 lists the findings of the measurement model assessment.

### 5.2. Structural Model

Before analyzing the direct effects of our PLS path model, we checked for collinearity issues among the constructs by evaluating the inner variance inflation factor (VIF) values of all pairs of endogenous and exogenous constructs. The VIF values ranged from 1.048 to 1.279, indicating no collinearity issues. We then proceeded to examine the regression results using bootstrapping with 5000 samples to assess the significance of path coefficients [56]. We used a one-tailed test to determine whether the influence of each exogenous construct on its corresponding endogenous construct was positive or negative [59].

Table 4 presents the results of our hypothesis testing for the direct effects. Our findings showed that hypotheses H1, H3, H4, and H6 were significantly supported as their confidence intervals did not include zero. However, hypotheses H2 and H5 were not supported due to their T-values. According to Hair et al. [56], the model’s predictive powers of attitude and intention were weak, considering the R^2^ values of 0.386 and 0.250 for the respective constructs.

### 5.3. Data Analysis Using a Data Multi-Analytical Approach Combining PLS-SEM and Artificial Neural Networks

Artificial neural networks can achieve high prediction accuracy, but they are not suitable for hypothesis testing because of their complex operation. On the other hand, conventional linear statistical techniques, for instance, PLS-SEM, only examine linear relationships, resulting in the over-simplification of complex decision-making processes in organizations [60]. Consequently, we integrated PLS-SEM and ANNs to build a multi-analytical approach to accurately determine the most significant predictors of attitude and intention. Several studies [61,62,63,64,65] have highlighted the effectiveness of this multi-analytical approach in providing in-depth and accurate insights for decision-makers. These valuable insights should help decision-makers to make well-informed decisions for achieving their desired outcomes in the respective research domains.

We developed our model using the feed-forward back-propagation algorithm, where the significant predictors of the PLS path model (perceived compatibility, perceived ease of use, and perceived usefulness) were considered the input neurons. Moreover, attitude was considered the dependent variable. Ten-fold cross-validation was considered to avoid over-fitting, using 90% of the data for network training and the remainder for testing [60]. The root mean square of errors (RMSEs) of training and testing for all ten ANN models were calculated to measure the models’ predictive accuracy. We examined the prediction accuracy of the models with Sigmoid as the activation function of all layers and two hidden layers because the models achieved a higher prediction accuracy with these settings. Figure 3 shows the ninth ANN model that achieved the lowest RMSE for testing. The influence of the input neuron on the output is determined by the color and width of the synaptic weight between them. The weight appears in blue when the influence is negative and in grey when it is positive.

Table 5 illustrates the RMSEs of the ten models. They achieved precise prediction accuracy since their average RMSEs were small (0.126 for training and testing). The RMSEs are in line with previous studies [66,67]. We conducted a sensitivity analysis to rank the predictors by determining their relative importance. Considering the normalized variable importance values in Table 6, the most significant predictor of attitude was perceived compatibility (100%), followed by perceived ease of use (66.73%), and perceived usefulness (36.10%). Since our findings showed that only attitude significantly influenced intention, we did not apply the multi-analytical approach to the intention construct.

## 6. Discussion

The original TAM’s external variables have not been studied adequately in examining user IT acceptance. Consequently, this model may not completely capture the complexity of real-world IT adoption and use. Therefore, we have proposed an extended TAM including two new predictors of attitude, namely, perceived compatibility, and perceived privacy and security issues, to study the acceptance of VHC among nurses. In our examination of the new factors, we found that nurses’ attitude toward using VHC was affected only by perceived compatibility. Our result aligns with Kuo et al. [30] who state that when mobile EMR systems are compatible with the work practices of nurses, this will increase their willingness to use these systems in their workplace. This finding is due to replacing paper-based forms currently completed by staff with computerized forms in VHC that simplify and speed up childhood immunization management. The system can also be installed and effortlessly operated on standard electronic devices in the workplace.

However, we could not determine any negative or significant relationship between perceived privacy and security issues and attitude, contradicting the findings of Jimma and Enyew [34] who pointed out that these issues negatively influenced the willingness of physicians and nurses to implement EMRs. VHC minimizes the recording of any sensitive records in the database, for instance, credit card information and confidential data about the parents and their children’s chronic diseases, to avoid causing any possible inconvenience to them in case of data leakage. Moreover, applying the latest encryption techniques to secure the stored data and requesting nurses to bypass the two-factor authentication to login into the system could probably inhibit any unauthorized access to the personal information of parents and infants. The effectiveness of these measures is further supported by Kruse et al. [68] who mentioned that utilizing security techniques could hinder unauthorized access to EHRs.

Evaluating the predominant factors of the TAM model, namely, perceived usefulness and perceived ease of use, demonstrated that they significantly improved nurses’ attitudes toward using VHC. These findings are supported by Alanazi et al. [33] and Tsai et al. [38]. Alanazi et al. [33] found that the perceptions of healthcare professionals regarding the adoption and use of EHR in Gulf Cooperation Council countries were influenced by the perceived usefulness and the perceived ease of use. Similarly, Tsai et al. [38] pointed out that these factors significantly influenced physicians’ attitudes toward EMRs. VHC simplifies the management of vaccination records for nurses by utilizing predefined and customizable immunization plans, allowing efficient and effortless childhood vaccination. Additionally, the system automatically sends SMS reminders to parents prior to vaccination appointments and recalls them if they miss an appointment. Furthermore, VHC provides visual reporting tools that allow nurses to monitor upcoming and overdue immunization appointments on a weekly, monthly, and annual basis, streamlining the process of monitoring vaccination coverage.

However, despite the significant relationship between perceived usefulness and attitude, this factor did not impact nurses’ intention to use VHC significantly, a finding supported by Gagnon et al. [35], who did not point out this significant relationship in the proposed psychosocial model when studying EHR acceptance by physicians. The use of text messaging vaccination reminder and recall systems, such as VHC, has not been widely explored in Malaysia’s healthcare sector, which primarily relies on paper-based systems to manage vaccinations. This observation is consistent with the results of our survey, demonstrating that a majority of the respondents had not previously used such systems in their workplaces. Consequently, their perceived usefulness of the system may not significantly increase their intentions, possibly due to a lack of expertise in utilizing the system. This assumption is supported by Kruse et al. [69], who identified a lack of user expertise as one of the barriers to EHR adoption. This insignificant finding may also be attributed to the extensive data entry required by nurses when using VHC for childhood immunization management. 

Nurses’ attitudes toward using VHC also had a significant and positive effect on their intentions, a finding supported by Kim et al. [39], who examined healthcare professionals’ adoption of mobile EMR using the unified theory of acceptance and use of technology (UTAUT). Furthermore, our finding is in line with the previous studies on hospital information technologies [21,22], and EMRs [38]. When nurses develop positive feelings about using VHC due to its compatibility with their current work practices, and its simplicity and helpfulness in the workplace, they would intend to use it in their healthcare institutions. Moreover, they do not express any significant concerns regarding the safety of children’s and parents’ personal information, as the system is highly secure and minimizes the collection of sensitive data.

Given that the attitude predictors, namely perceived compatibility, perceived ease of use, and perceived usefulness were found to be significant, an artificial neural network was utilized to determine which predictors had a greater impact on attitude. Based on the ANN models, perceived compatibility was the most significant factor of attitude, followed by perceived ease of use and perceived usefulness. When VHC is perceived to be compatible, it is easier for nurses to integrate it into their daily work routines, leading to a more positive attitude toward its use. Then, when the system is perceived as easy to use, nurses tend to embrace and use it regularly, improving their attitude toward using the system. Next, nurses are more likely to recognize the value and benefits of VHC when perceive it as useful, leading to a positive perception of the system. Figure 4 demonstrates the significant and insignificant effects of the final research model.

## 7. Contributions

Our study offers the following practical and theoretical contributions:

### 7.1. Practical Contributions

Our findings show that a text messaging vaccination reminder and recall system like VHC should simplify immunization administration to be highly accepted by nurses in the workplace. Therefore, considering the inclusion of innovative features like optical character recognition (OCR) and speech recognition to facilitate tedious data entry tasks could increase their positive perceptions of the system. In addition to using SMS as the primary communication channel, incorporating automated phone calls can be beneficial for parents with impaired vision. This additional feature can improve accessibility and ensure that all parents receive important immunization reminders. Furthermore, utilizing email-based reminders can enable nurses to provide more comprehensive educational information to parents, emphasizing the significance of immunization and addressing any potential concerns or questions they may have.

Moreover, the system’s process and design should represent nurses’ current work style precisely to be compatible with healthcare organizations. Consequently, software developers are encouraged to entirely comprehend the existing paper-based systems by interviewing the board of directors and nurses and assessing vaccination record forms to effectively implement them in the system. They are also encouraged to develop several prototypes, perform usability tests, and seek feedback from the board of directors and nurses to develop a compatible system that addresses the requirements of healthcare institutions. Meanwhile, the board of directors and nurses need to allocate sufficient time for software developers to explain their current vaccination procedures and requirements.

In addition, software developers should make sure that VHC is useful so that nurses feel positive about it. This means that the system should be designed in a way that makes it easy for nurses to find the information they need to complete their tasks efficiently. The system should also provide nurses with the tools they need to deliver high-quality care for registered infants. Finally, the board of directors and software developers should organize training workshops for nurses to help them learn how to use VHC efficiently.

The results of the multi-analytical approach using PLS-SEM and ANN also showed that VHC developers should focus on making the system compatible, easy to use, and useful in the workplace to improve the attitude of nurses toward VHC. 

### 7.2. Theoretical Contribution

This study contributes to the limited literature on the acceptance of VHC and text messaging vaccination reminder and recall systems among healthcare institutions. Since the external predictors of the original TAM have not been studied in sufficient depth, we proposed new predictors of attitude, namely, perceived compatibility and perceived privacy and security issues. Among those, we found that perceived compatibility led to the positive perspectives of nurses on the system, adding to the present predictors of attitude in the TAM. Our findings also showed that the predominant factors of the TAM, namely, perceived usefulness and perceived ease of use, significantly improved nurses’ attitudes toward using VHC. Their attitude toward using VHC also led to their determination to use the system. Finally, applying a multi-analytical approach of PLS-SEM and ANN helped us determine the most significant factors of nurses’ attitudes with higher accuracy. We found that perceived compatibility was the most significant factor influencing nurses’ attitudes towards using VHC, followed by perceived ease of use and perceived usefulness.

## 8. Conclusions

Despite the potential benefits of text messaging vaccination reminder and recall systems in simplifying and accelerating childhood immunization administration, their use has not been widely explored in Malaysia’s healthcare sector. Therefore, we conducted a survey to examine the factors that influence nurses’ attitudes and intentions to use VHC within the healthcare sector. Considering the external variables of TAM have not been clearly determined, we proposed new predictors of attitude, namely, perceived compatibility, and perceived privacy and security issues, to address the limitations of TAM. By including these additional predictors, we aimed to provide a more comprehensive understanding of the factors that influence technology acceptance. Our findings demonstrate that among the new predictors, only perceived compatibility could improve the nurses’ attitude, demonstrating that the compatibility of VHC with the nurses’ current work setting had a vital role in developing their positive perspective on the system. Furthermore, we noticed that the nurses would be optimistic about the use of VHC if they perceived it as useful and easy to use in their workplace. Their attitudes toward using VHC also played a determining role in their intention to use the system. 

The findings of the multi-analytical approach combining PLS-SEM and ANN also showed that perceived compatibility was the most significant factor influencing nurses’ attitudes, followed by perceived ease of use and perceived usefulness. These insights may help healthcare decision-makers in making informed decisions that advance health services. Our findings may have significant implications for managerial and software development practices within the healthcare industry, particularly with regard to the successful development, implementation, and acceptance of VHC and similar systems, which may ultimately lead to raising childhood vaccination coverage and protecting infants against preventable diseases. Moreover, they could further contribute to the currently limited literature on the acceptance factors of these systems and mitigate the limitations of TAM by providing further knowledge about the predictors driving positive attitudes toward the systems. 

Despite the importance of this research, future studies should be conducted in the private sector to assess the acceptance of VHC in this setting, which may have different needs and priorities than the public sector. This in turn would improve the generalizability of our findings. Moreover, the research model was only evaluated by collecting nurses’ responses. Therefore, considering other medical personnel with a high possibility of using VHC, such as physicians might generalize the results. Furthermore, the data were collected using self-administrated questionnaires, which might cause ambiguity for the respondents and encourage them to answer the measurement items in a socially desirable manner. Therefore, applying mixed research methods, including surveys and interviews, could be considered to overcome these issues. Finally, trial studies should be conducted to evaluate the effectiveness of VHC in improving the completion and timeliness of immunizations among infants.

## Figures and Tables

**Figure 1 vaccines-11-01331-f001:**
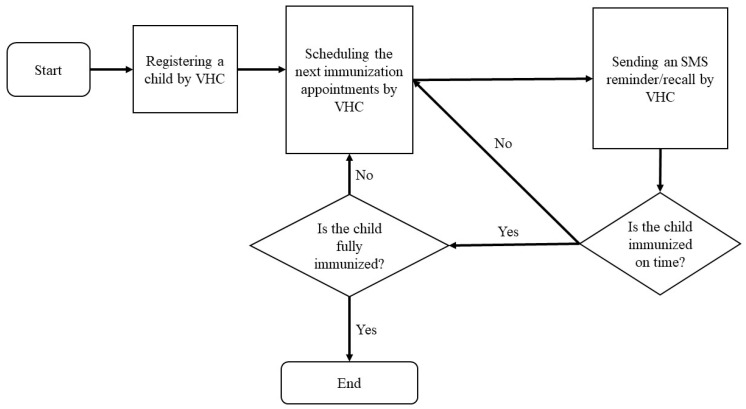
The Vaccination Schedule Processes by VHC.

**Figure 2 vaccines-11-01331-f002:**
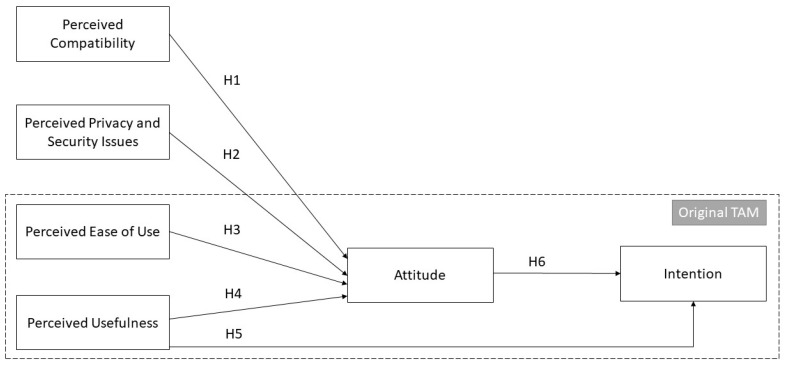
The proposed research model. Note. Hypotheses H1 to H6 represent the direct relationships.

**Figure 3 vaccines-11-01331-f003:**
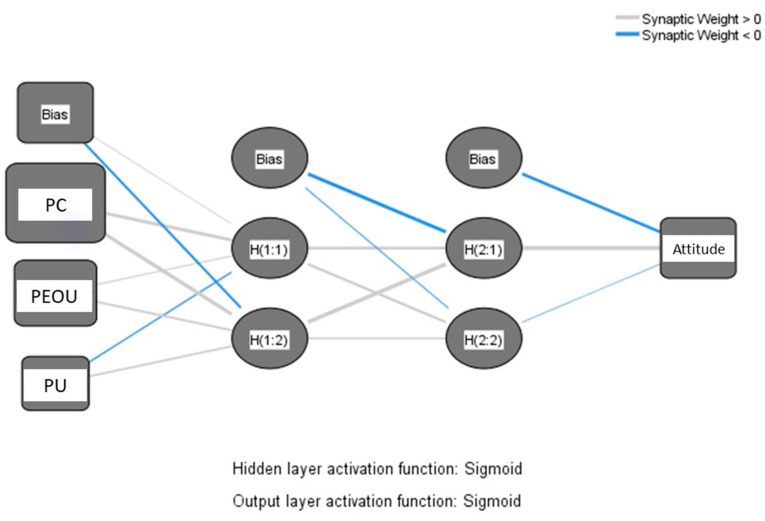
The ninth neural network model. Note: PC: perceived compatibility; PEOU: perceived ease of use; PU: perceived usefulness.

**Figure 4 vaccines-11-01331-f004:**
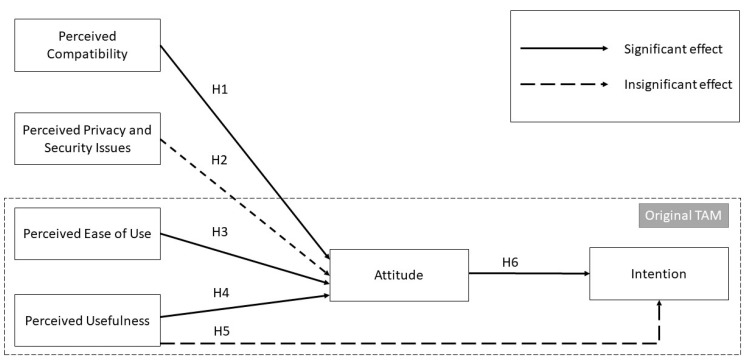
The final research model.

**Table 1 vaccines-11-01331-t001:** Measurement items of constructs.

Construct	Items	References
Perceived compatibility	a. Using VHC is compatible with all aspects of our work. b. Using VHC is entirely compatible with our current situation. c. We think using VHC fits adequately with the way we like to work. d. We think using VHC fits into our work style.	[40]
Perceived privacy and security issues	a. Using VHC would lead to a loss of patients’ information privacy since the information collected could be used without our knowledge. b. Using VHC would lead to a loss of patients’ information security since the information collected could be used without our knowledge. c. We would be concerned about patients’ information security when using VHC. d. We would be concerned about patients’ information privacy when using VHC.	[32,41]
Perceived ease of use	a. Our interaction with VHC would be clear and understandable. b. Interacting with VHC would not demand a lot of our mental effort. c. We would find it easy to get VHC to do what we want it to do. d. We would find VHC to be easy to use.	[32]
Perceived usefulness	a. Using VHC would improve our effectiveness in our job. b. Using VHC would improve our job performance. c. Using VHC would make it easier to perform our job. d. We would find VHC to be useful in our job.	[32]
Attitude toward using VHC	a. Using VHC is beneficial for us. b. Using VHC is a good idea. c. Working with VHC is pleasant. d. Overall, our attitude toward using VHC is positive.	[32,42]
Intention to use VHC	a. We intend to use VHC in the future. b. We predict we would use VHC in the future. c. We plan to use VHC in the future. d. Assuming we had access to VHC, we would intend to use it.	[32,43]

**Table 2 vaccines-11-01331-t002:** The demographic profile of the respondents.

Demographics	Frequency	Percentage
Age		
below 18–24 years	28	23.1
25–34 years	40	33.1
35–44 years	24	19.8
45–54 years	17	14.0
55–64 years	7	5.8
65 years or above	5	4.1
Gender		
Male	39	32.2
Female	82	67.8
Education		
Secondary school	44	36.4
Diploma	50	41.3
Bachelor	16	13.2
Master	11	9.1
Working Experience		
Below 3 years	41	33.9
3 to 5 years	53	43.8
5 to 10 years	14	11.6
Above 10 years	13	10.7
Prior Acceptance of a Text Messaging Vaccination Reminder and Recall System		
Yes	30	24.8
No	91	75.2

**Table 3 vaccines-11-01331-t003:** Reflective measurement model assessment.

Construct	Number of Items	Deleted Items	Factor Loading	CR	AVE	HTMT Confidence Interval Excluded 1
PC	4	None	0.788	0.901	0.695	Yes
			0.850			
			0.855			
			0.840			
PPSI	4	2		0.832	0.714	Yes
			0.776			
			0.909			
PEOU	4	None		0.870	0.628	Yes
			0.695			
			0.884			
			0.817			
			0.761			
PU	4	None		0.896	0.685	Yes
			0.782			
			0.853			
			0.766			
			0.901			
Attitude	4	None		0.822	0.536	Yes
			0.752			
			0.741			
			0.718			
			0.716			
Intention	4	1		0.839	0.636	Yes
			0.825			
			0.812			
			0.753			

Note. PC: perceived compatibility; PPSI: perceived privacy and security issues; PEOU: perceived ease of use; PU: perceived usefulness; CR: composite reliability; AVE: average variance extracted; HTMT: heterotrait–monotrait ratio.

**Table 4 vaccines-11-01331-t004:** Hypotheses testing for direct effects.

Path (Hypothesis)	Path Coefficient	T-Value	*p*-Value	95% Confidence Intervals	Significance (*p* < 0.05)
H1: PC → Attitude	0.474 **	6.713	0.000	[0.357, 0.589]	Yes
H2: PPSI → Attitude	0.061	0.823	0.205	[−0.056, 0.184]	No
H3: PEOU → Attitude	0.178 *	2.314	0.010	[0.062, 0.313]	Yes
H4: PU → Attitude	0.109 *	1.685	0.046	[0.009, 0.217]	Yes
H5: PU → Intention	0.095	0.937	0.174	[−0.069, 0.252]	No
H6: Attitude → Intention	0.471 **	5.904	0.000	[0.342, 0.604]	Yes

Note: ** Significance at *p* < 0.01, * Significance at *p* < 0.05. PC: perceived compatibility; PPSI: perceived privacy and security issues; PEOU: perceived ease of use; PU: perceived usefulness.

**Table 5 vaccines-11-01331-t005:** Root mean square of errors (RMSEs) of the neural network models.

Network	Training	Testing
ANN1	0.131	0.230
ANN2	0.136	0.118
ANN3	0.125	0.124
ANN4	0.125	0.113
ANN5	0.127	0.093
ANN6	0.129	0.126
ANN7	0.124	0.108
ANN8	0.123	0.116
ANN9	0.125	0.090
ANN10	0.119	0.143
Mean	0.126	0.126
Standard Deviation	0.004	0.039

Note: ANN: artificial neural network.

**Table 6 vaccines-11-01331-t006:** Independent variable importance.

Network	PC	PEOU	PU
1	0.431	0.366	0.202
2	0.563	0.284	0.154
3	0.489	0.354	0.157
4	0.504	0.284	0.213
5	0.514	0.332	0.154
6	0.444	0.398	0.159
7	0.451	0.347	0.202
8	0.560	0.254	0.186
9	0.509	0.327	0.164
10	0.469	0.343	0.189
Average	0.493	0.329	0.178
The Normalized Variable Importance	100.000	66.734	36.105

Note: PC: perceived compatibility; PEOU: perceived ease of use; PU: perceived usefulness.

## Data Availability

The dataset can be provided by the authors.

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
