# Peer review of "Acceptance of a Text Messaging Vaccination Reminder and Recall System in Malaysia’s Healthcare Sector: Extending the Technology Acceptance Model"

_vaccines, 2023, doi:10.3390/vaccines11081331_

Round 1

Reviewer 1 Report

In this manuscript titled "Acceptance of a Text Messaging Vaccination Reminder and Recall System in Malaysia's Healthcare Sector: Extending the Technology Acceptance Model," the authors address the issue of ineffective paper-based vaccination systems in Malaysian healthcare institutions and propose a text messaging vaccination reminder and recall system called Virtual Health Connect (VHC) as a solution. They investigate factors influencing nurses' attitudes and intentions to use VHC using an extended version of the Technology Acceptance Model (TAM). Overall, the manuscript presents an interesting study that addresses a relevant issue in healthcare. However, there are a few areas that require attention and improvement:

1. The introduction should provide more background information on the current state of vaccination systems in Malaysian healthcare institutions. It is important to clearly establish the significance and urgency of the problem, such as the prevalence of missed appointments, incomplete vaccinations, and outbreaks of preventable diseases among infants.

2. The introduction section requires further elaboration to provide a logical stance that motivates the study. While the topic is introduced, it would greatly benefit from a more comprehensive discussion of the problem statement, its significance, and the research gap this study aims to fill. This will help readers better understand the relevance and importance of the research question being addressed.

3. It would be beneficial to strengthen the justification for using the Technology Acceptance Model (TAM) in this study, considering that TAM is considered by some as an outdated model in technology acceptance research. Please refer to the following publication for insights: https://doi.org/10.1007/978-3-030-64987-6_1

4. The authors conducted a survey among 121 nurses in Malaysian government hospitals and clinics to test their extended TAM model. While the sample size is mentioned, it would be beneficial to provide additional details regarding the sampling method and any potential biases. Moreover, a discussion on the limitations of the study, such as generalizability and potential response bias, should be included.

5. To justify the combined use of partial least squares-structural equation modelling (PLS-SEM) and artificial neural network (ANN) in the analysis, it is recommended to include a subsection titled "Data Analysis" in the methodology section. This will provide a detailed explanation for the selection of PLS-SEM and ANN as appropriate analysis techniques. Please refer to the following references for justifying the selection of PLS-SEM and ANN:

Suggested References:

  1. https://doi.org/10.1108/IJBM-05-2022-0202]
  2. [ https://doi.org/10.1016/j.heliyon.2023.e16299]
  3.  https://doi.org/10.1108/JPBM-02-2022-3855]
  4.  https://doi.org/10.1080/10494820.2022.2075014]
  5.  https://doi.org/10.3390/su141912726]

6. To improve the structure of the manuscript, it is recommended to move the "Research Limitations and Future Directions" section under the "Conclusion" section. This will allow for a more comprehensive conclusion that encompasses the study's findings, limitations, and future research directions. This adjustment will enhance the overall clarity and cohesion of the manuscript.

Moderate editing of English language required

Author Response

Dear Reviewer,

We would like to express our sincere gratitude for your valuable insights and comments on our manuscript. Your feedback has been invaluable in helping us improve the quality of our work, and we are wholeheartedly grateful for them.

Please kindly be informed that your important feedback has been carefully addressed and mainly highlighted in turquoise, and we have made the necessary revisions to improve the quality of our work. We have also thoroughly checked our manuscript for proofreading.

Please feel free to inform us if you believe that further improvements are required. We value your expertise and appreciate your contribution to the development of our manuscript.

Thank you once again for your precious time and valuable effort. We look forward to hearing from you.

Yours faithfully,

 Dr. Kamal Karkonasasi

Kindly be informed that we have carefully responded to your significant and valuable feedback as mentioned below:

1. The introduction should provide more background information on the current state of vaccination systems in Malaysian healthcare institutions. It is important to clearly establish the significance and urgency of the problem, such as the prevalence of missed appointments, incomplete vaccinations, and outbreaks of preventable diseases among infants.

Thanks so much for your valuable insights. We have offered the reader more detailed background information on the current state of the vaccination system in Malaysia by comparing the country’s childhood immunization rates with the global average. We have also pointed out the outbreaks of some vaccine-preventable diseases like tuberculosis are because of declining immunization rates in Malaysia in recent years which clearly establishes the significance and urgency of the problem. We have added the following paragraph in “1. Introduction” (Page 1, Lines 39 to 48):

“According to recent data published in 2022 by the World Health Organization (WHO), Malaysia’s childhood immunization rates are higher than the global average. For example, the BCG immunization rate among infants in Malaysia was 97.66%, compared to the global average of 84%. However, immunization rates against vaccine-preventable diseases have decreased in Malaysia in recent years, particularly for follow-up doses (World Health Organization, n.d.). This has led to an outbreak of diseases like tuberculosis. In 2022, there were 25,391 reported tuberculosis cases, compared to 21,727 a year earlier - an increase of 17%. The number of deaths also increased by 12%, from 2,288 to 2,572 during the same period (CGTN, 2023).”  

2. The introduction section requires further elaboration to provide a logical stance that motivates the study. While the topic is introduced, it would greatly benefit from a more comprehensive discussion of the problem statement, its significance, and the research gap this study aims to fill. This will help readers better understand the relevance and importance of the research question being addressed.

We are highly grateful for your important comments. We have thoroughly improved “1. Introduction” by offering the reader a detailed discussion of the problem statement, its importance, and the research gap we tend to address. We have highlighted the modifications in turquoise and bright green. Please feel free to inform us if you think that further improvements are needed. We highly appreciate that

3. It would be beneficial to strengthen the justification for using the Technology Acceptance Model (TAM) in this study, considering that TAM is considered by some as an outdated model in technology acceptance research. Please refer to the following publication for insights: https://doi.org/10.1007/978-3-030-64987-6_1

We are wholeheartedly grateful for your insights. Indeed, we found them highly relevant and helpful to back up our underlying model. Kindly be informed that we have added the following paragraph at the beginning of “2. The Proposed Research Model” (Page 4, Lines 138 to 142):

”The TAM, proposed by Davis (1989), is a widely recognized model for analyzing the factors that influence user behavior and use of numerous technologies. Despite being proposed over three decades ago, the model remains relevant and is still being applied, modified, and extended by researchers in several domains, including healthcare (Al-Emran & Granić, 2021).”

4. The authors conducted a survey among 121 nurses in Malaysian government hospitals and clinics to test their extended TAM model. a) While the sample size is mentioned, it would be beneficial to provide additional details regarding the sampling method and any potential biases. b) Moreover, a discussion on the limitations of the study, such as generalizability and potential response bias, should be included.

Thanks a lot. For part a, we have added further details to “4.1. Survey Design” (pages 6 to 8) and “4.2. Sampling and Data Collection Procedures” (page 8) which are highlighted in bright green for your kind information. For part b, we have discussed the limitations of the study, for instance, generalizability and potential response bias in “8. Conclusion” (page 17). We have also provided some insights in this section to overcome these limitations (pages 17 and 18, lines 567 to 578). Please do not hesitate to inform us if we could address your valuable comments precisely. Thanks again.

5. To justify the combined use of partial least squares-structural equation modeling (PLS-SEM) and artificial neural network (ANN) in the analysis, it is recommended to include a subsection titled "Data Analysis" in the methodology section. This will provide a detailed explanation for the selection of PLS-SEM and ANN as appropriate analysis techniques. Please refer to the following references for justifying the selection of PLS-SEM and ANN:

Suggested References:

  1. https://doi.org/10.1108/IJBM-05-2022-0202
  2. https://doi.org/10.1016/j.heliyon.2023.e16299
  3.  https://doi.org/10.1108/JPBM-02-2022-3855
  4.  https://doi.org/10.1080/10494820.2022.2075014
  5.  https://doi.org/10.3390/su141912726

Thanks so much for your important insights. To address your valuable comment and justify our data analysis approach, in “5.3. Data Analysis Using Data Multi-Analytical Approach of PLS-SEM and Artificial Neural Network” (page 12, lines 383 to 387), we have mentioned as follows:

 “Several studies (Al-Sharafi et al. 2022, 2023; Dang et al., 2023; Mohd Rahim et al., 2022; Theadora et al., 2023) have highlighted the effectiveness of this multi-analytical approach in providing in-depth and accurate insights for decision-makers. These valuable insights shall help decision-makers to perform well-informed decisions for achieving their desired outcomes in the respective research domain.”

6. To improve the structure of the manuscript, it is recommended to move the "Research Limitations and Future Directions" section under the "Conclusion" section. This will allow for a more comprehensive conclusion that encompasses the study's findings, limitations, and future research directions. This adjustment will enhance the overall clarity and cohesion of the manuscript.

Thank you a lot. We have added the “Research Limitations and Future Directions" section under "8. Conclusion" while preserving the flow (pages 17 to 18, lines 567 to 578). 

Reviewer 2 Report

Strengths

1.     The study proposes Virtual Health Connect (VHC), a text messaging vaccination reminder and recall system, to replace the current inefficient paper-based systems in Malaysian healthcare sector that might lead to improving the completion and timeliness of immunizations among children. The proposed model examines the factors affecting nurses’ attitude and intention to use the system.

2.     The study analysed data using partial least squares-structural equation modeling (PLS-SEM) to examine the significant 22 factors influencing nurses’ attitude and intention to use Virtual Health Connect. Moreover, we applied artificial neural 23 network (ANN) to determine the most significant factors of acceptance with higher accuracy.

3.     The study has been well conceived and articulated by the authors. The data collection methods, data analysis and interpretations are well presented.

4.     The study also presented the practical contributions of the research in the Malaysian health sector and its theoretical implications.

5.     The conclusions are emerged from the analysis of the study data and discussions

Weaknesses

1.     Malaysia’s achievement of immunization agenda 2030 and coverage targets, key challenges and strategies should be highlighted in the introductory section.

2.     Line 40-41 may be substantiated with data from the WHO or Malaysian Ministry of Health data

3.     Period of data collection, language used to develop questionnaire should be highlighted in the methodology section.

4.     Justification for conducting the study in Northern region of Malaysia should be provided. Can the results be generalized.

5.     It is not clear whether questionnaires were sent to all government hospitals and clinics in Northern region of Malaysia. What are the total number of government hospitals and clinics in Northern region of Malaysia?

6.     Methodology section should also highlight the details of basis for development of questionnaire, details of items included in each construct, whether questionnaire already validated?

7.     What about ethical considerations? Is the study approved by any professional ethical body? Were respondents informed about the ethical issues involved in the study?

8.     Study claims that a survey among 121 nurses in Malaysian government hospitals and clinics was conducted to test the model. However, methodology we distributed the self-administrated questionnaires by post among 452 government hospitals and clinics. It is not clear each questionnaire represent nurse as a respondent or represents hospitals? How the nurses were selected within hospitals. Please clarify.

9.     Discussion section may be enriched with inclusion of findings from other studies. For example, the findings show that there is insignificant relationship between perceived privacy and security issues and attitude of nurses. These findings should be discussed in view of other study findings.

Only minor edition required

Author Response

Dear Reviewer,

We would like to take this opportunity to thank you for your thorough review and insightful comments on our manuscript. Your feedback has been invaluable in helping us improve the quality of our work.

We have carefully addressed your comments and made the necessary revisions mainly highlighted in bright green. If you believe that further improvements are required, please do not hesitate to let us know. We value your expertise and appreciate your contribution to the development of our manuscript.

Thank you once again for your valuable time and priceless effort in reviewing our study. We look forward to hearing from you.

Yours faithfully,

 Dr. Kamal Karkonasasi

--

Kindly be informed that we have carefully responded to your significant and valuable feedback as mentioned below:

1. Malaysia’s achievement of immunization agenda 2030 and coverage targets, key challenges and strategies should be highlighted in the introductory section.

We have added the following to examine Malaysia’s achievement of IA2030 and coverage targets, key challenges and strategies in “1. introduction” (page 2, lines 49 to 56):

“Consequently, the Malaysia healthcare system still lags behind the WHO’s Immunization Agenda 2030 (IA2030) that envisions “a world where everyone, everywhere, at every age, fully benefits from vaccines to improve health and well-being” (World Health Organization, 2020). The recent decreases in immunization rates and corresponding increases in vaccine-preventable diseases like tuberculosis demonstrate that Malaysia has yet to fully align with WHO goals for comprehensive immunization coverage. More efforts are needed to boost immunization rates and ensure the infants are protected against contagious illnesses.“

2. Lines 40-41 may be substantiated with data from the WHO or Malaysian Ministry of Health data.

Thank you. We have further supported the previous lines 40 to 41 in “1. Introduction” by adding more information from the WHO and MOH to demonstrate the current state of the vaccination system and healthcare system in Malaysia as follows (Highlighted in blue, pages 1 and 2, lines 39 to 48):

“According to recent data published in 2022 by the World Health Organization (WHO), Malaysia’s childhood immunization rates are higher than the global average. For example, the BCG immunization rate among infants in Malaysia was 97.66%, compared to the global average of 84%. However, immunization rates against vaccine-preventable diseases have decreased in Malaysia in recent years (World Health Organization, n.d.). This has led to an outbreak of diseases like tuberculosis. In 2022, there were 25,391 reported tuberculosis cases, compared to 21,727 a year earlier - an increase of 17%. The number of deaths also increased by 12%, from 2,288 to 2,572 during the same period (CGTN, 2023).”  

3. Period of data collection, language used to develop the questionnaire should be highlighted in the methodology section. 

We are highly grateful for your insights. The period of data collection was from September 2022 to April 2023 as mentioned in “4.2. Sampling and Data Collection Procedures” (page 8, lines 275 to 276). Moreover, the survey was conducted in the English language as mentioned in “4.2. Sampling and Data Collection Procedures” (page 8, line 270).

4. Justification for conducting the study in Northern region of Malaysia should be provided. Can the results be generalized?

Thanks a lot for bringing this matter to our attention. We sincerely apologize for any inconvenience caused. Indeed, we have conducted our survey in Peninsular Malaysia, i.e. excluding Sabah and Sarawak. We have justified our decision and explained why we believe our findings still may have a high degree of generalizability in “4.2. Sampling and Data Collection Procedures” (page 8, lines 273 to 276) partially mentioned below:

“Given the similarities between government hospitals and clinics in Malaysia, it can be inferred that our findings may have a high degree of generalizability”. We hope that we could address your concern. Thanks again.

5. It is not clear whether questionnaires were sent to all government hospitals and clinics in Northern region of Malaysia. What is the total number of government hospitals and clinics in Northern region of Malaysia?

Thank you for your valuable insights. Kindly take note that we have conducted our survey in Peninsular Malaysia, i.e. excluding Sabah and Sarawak. Once again, we apologize for any inconvenience this may have caused. 

According to Malaysia P.R.K.K. (n.d.), Peninsular Malaysia consists of 93 government hospitals and 2,411 government clinics, as mentioned in “4.2. Sampling and Data Collection Procedures” (page 8, lines 261 to 262. Among these, we have only selected 452 government hospitals and clinics using a simple random sampling technique. Moreover, we did not send our surveys to all government hospitals and clinics in Peninsular Malaysia due to time and budget constraints However, considering the similarities between government hospitals and clinics in Malaysia, we can infer that our findings may be highly generalizable.

6. Methodology section should highlight the details of basis for development of questionnaire details of items included in each construct, and whether the questionnaire has already been validated.

Thanks for your valuable insights. In order to address your important comment, we have added the following: “Participation in the survey was voluntary, and we assured participants that their data would remain confidential and be used only for academic purposes.” (pages 6 to 7, lines 237 to 239) We have also added Table 1. Measurement Items of Constructs (pages 7 to 8) in which we have demonstrated the measurement items of each construct with their references. The questionnaire has been validated as explained in the following: we have requested two experts from Universiti Sains Malaysia to review the content validity of our survey as mentioned in “4.1. Survey Design” (page 8, lines 247 to 249). Moreover, in “5.1. Measurement Model” (pages 10 to 11), we have assessed the reflective measurement model’s convergent and discriminant validity using confirmatory factor analysis (CFA) in SmartPLS 4 software (Ringle et al., 2022). Convergent validity was evaluated through factor loading, composite reliability (CR), and average variance extracted (AVE) (Chang, 2020). Furthermore, we have examined discriminant validity of the measurement model using Heterotrait–Monotrait Ratio (HTMT). 

7. What about ethical considerations? Is the study approved by any professional ethical body? Were respondents informed about the ethical issues involved in the study?

Thanks for your important insights. To address your valuable comment, in “Institutional Review Board Statement” (page 18, lines 591 to 593), we have mentioned the following: “The study was conducted according to the guidelines of the Declaration of Helsinki and approved by the Human Research Ethics Committee from Universiti Sains Malaysia (FWA Reg. NO: 00007718; IRB Reg. NO: 00004494).”

Moreover, the participants are informed about the ethical issues as stated in “4.1. Survey Design” (pages 6 to 7, lines 237 to 239): “Participation in the survey was voluntary, and we assured participants that their data would remain confidential and be used only for academic purposes.

8. Study claims that a survey among 121 nurses in Malaysian government hospitals and clinics was conducted to test the model. However, methodology we distributed the self-administrated questionnaires by post among 452 government hospitals and clinics. It is not clear whether each questionnaire represent nurse as a respondent or represents hospitals? How the nurses were selected within hospital? Please clarify.

Thanks so much for your fruitful comment. Kindly be informed that “Each nurse who participated in the survey represented a government hospital or clinic since we aimed to study the acceptance of VHC at an organization level” as mentioned in “4.1. Survey Design” (page 6, lines 230 to 233). For your kind information, the criteria for selecting nurses are mentioned in “4.1. Survey Design” (page 6, lines 225 to 227): “In the introduction of the survey, we explained that the survey should be completed by a randomly selected nurse who is competent in English and IT literacy.”

9. Discussion section may be enriched with inclusion of findings from other studies. For example, the findings show that there is insignificant relationship between perceived privacy and security issues and attitude of nurses. These findings should be discussed in view of other study findings.

Thank you very much for your valuable insights. We have thoroughly enriched our discussions on pages 14 to 16 from the findings of the other studies. Moreover, we have added new references: Kruse et al. (2016, 2017), to further expand and support our discussions. We hope that we have addressed your valuable comment precisely. Please do not hesitate to inform us if we could further improve this section. Thanks once again.

Reviewer 3 Report

See attached file

good quality

Author Response

Dear Reviewer,

We would like to express our sincere gratitude for your valuable insights and positive comments on our manuscript. Your feedback has been invaluable in helping us improve the quality of our work, and we are wholeheartedly grateful for them.

Please kindly be informed that your important feedback has been carefully addressed and mainly highlighted in turquoise and bright green, and we have made the necessary revisions to improve the quality of our work. We have also thoroughly checked our manuscript for proofreading.

Please feel free to inform us if you believe that further improvements are required. We value your expertise and appreciate your contribution to the development of our manuscript.

Thank you once again for your positive feedback. We look forward to hearing from you.

Yours faithfully,

 Dr. Kamal Karkonasasi

Round 2

Reviewer 1 Report

The authors have addressed all comments from the previous review. The manuscript is now improved and ready for publication.

I suggest a final language check to ensure clarity and precision before publication.